# Mediating Effects of Discipline Approaches on the Relationship between Parental Mental Health and Adolescent Antisocial Behaviours: Retrospective Study of a Multisystemic Therapy Intervention

**DOI:** 10.3390/ijerph192013418

**Published:** 2022-10-17

**Authors:** Leartluk Nuntavisit, Mark Porter

**Affiliations:** Multisystemic Therapy Program, Specialised Child and Adolescent Mental Health Service (CAMHS), Department of Health, Fremantle Hospital, Fremantle, WA 6160, Australia

**Keywords:** Multisystemic Therapy, conduct disorder, antisocial behaviour, adolescents, parental mental health, discipline approaches

## Abstract

Poor parental mental health is one of the risk factors for child emotional and behavioural problems because it reduces caregiver’s ability to provide appropriate care for their child. This study aimed to measure changes in parenting factors and adolescent behaviours after Multisystemic Therapy (MST), and to explore the mediating role of discipline approaches on the relationship between parental mental health and adolescent behavioural problems. This retrospective study extracted data collected from 193 families engaged with the MST research program during 2014–2019. Data was collected at different time points (pre-treatment, post-treatment, 6- and 12-months follow-up). Statistically significant changes were found in adolescent behaviours and parenting factors following the MST intervention and these positive changes were maintained over the following 12 months. Results of the parallel multiple mediator model analysis confirmed mediating effects of discipline approaches on the relationship between parental mental health and adolescent’s behavioural problems. The findings suggested that parental mental well-being significantly contributes to effectiveness of parenting, which resulted in positive changes in adolescent’s behavioural problems. It is recommended caregiver’s parental skills and any mental health issues are addressed during the intervention to enhance positive outcomes in adolescent behaviour.

## 1. Introduction

Oppositional defiant disorder (ODD) and conduct disorder (CD) are one of the most common mental and behavioral problems in children [1]. A meta-analysis conducted in 2015 by Polanczyk et al. [2] reported that disruptive disorders was the second highest prevalence of mental disorders in children and adolescent at 5.7% (the highest prevalence was anxiety disorder at 6.5%). A report from the Mental Health of Australian Children and Adolescents Survey [3] indicated that approximately 8% of all Australian children and adolescents met diagnostic criteria for oppositional defiant disorder or conduct disorder. In addition, almost half of these children and adolescents reportedly had co-occurring mental disorders, e.g., ADHD and mood disorders. Children with oppositional problems are negativistic, hostile and defiant, and if untreated, they often develop a conduct disorder exhibiting a range of delinquent behaviours including bullying, physical fights, deliberately destroying other’s property, breaking into properties or cars, staying out late at night without permission, substance use, and absconding from home and school. Without effective intervention, conduct disorder is a reliable predictor of various adult mental illness, substance abuse, chronic unemployment, domestic violence and incarceration [4]. The finding of a systemic review on longitudinal studies investigating childhood factors contributing to domestic violence in adulthood indicated that child abuse/neglect, adversity in the family of origin, child/adolescent aggressive behaviours, substance use and negative peer influences were the strongest predictors of domestic violence perpetration and victimization in adulthood [5] Many of these difficulties are very high cost to families and the wider community. In order to decrease the likelihood of these negative life trajectories, factors that influence children’s emotional and behavioural problems need to be identified and integrated into interventions.

The role of parent–child interaction is a significant contributor to children’s emotional and behavioural well-being that has been highlighted in several theoretical works [6,7]. Enhancing interpersonal relationship and supports within children’s social ecology, was found to be key to maintaining desired outcomes in children such as improved physical and mental health, cognitive development and educational attainment [8,9]. Therefore, a common goal for effective interventions for children and adolescent with problem behaviours is to support caregivers to implement effective parenting, because this has been shown to improve the wellbeing of children and reduce impacts of social disadvantage [10,11]. Behavioural parenting training is an evidence-based intervention based on the social interactional model explaining the vital role of parenting behaviours on children and adolescent disruptive behaviours [12]. BPT has been integrated into many treatment programs for children and adolescents with disruptive behaviours. In this model, positive changes in parenting behaviours were the key to reducing disruptive behaviours. The positive changes in parenting skills include: reducing harsh punishments or criticism; using positive reinforcements for appropriate behaviours; and increasing supervision and monitoring behaviour. Reduction of negative parenting, e.g., hostility, lack of rules, physical punishment were found to be the strongest predictor in improved child problem behaviours [13]. The systemic review of BPT [14] noted that many treatment programs that implement BPT as a main intervention reported positive children outcomes; however it seems to be more effective with younger children compared with adolescents with more severe problems.

Children with high risk for antisocial behaviours and mental health disorders have often experienced childhood trauma, poverty, family disruptions and abuse [15,16], and are more likely to have caregivers with various difficulties including substance misuse, domestic violence, and chronic engagement with adult mental health services [17]. Australian research examining the relationship between family’s socioeconomic status and aggression among school-age children, found that students experiencing socioeconomic disadvantage, e.g., poverty, parent’s lack of formal education and unemployment are more likely to have poor academic performance and exhibit aggressive behaviours in the school setting [18]. These family adversities were found to impact on the caregiver’s ability to parent effectively [19]. Several studies have shown parent mental well-being was a key contributor to effective parenting [11,20,21]. Lack of effective parenting has been associated with adolescent mental health problems, substance misuse, school disengagement, and juvenile offending [22,23]. Effective parenting requires presence of caregiver warmth and lack of hostility, clear rules and expectations, and consistent interest in the child’s life [24,25]. Therefore, interventions which aim to improve parent–child interaction should identify the protective factors such as caregiver’s mental well-being as this will affect the caregiver’s ability to provide care for children. The relationship between parent’s mental health and parenting can be explained by a process model of parental functioning proposed by Belsky, J. The model explains that there are three determinants of parenting: (1) personal psychological resources of parents, (2) characteristics of the child and (3) contextual sources of stress and support [26]. The model proposes that parental psychological wellbeing promotes competent parenting; however, the source of contextual stress, e.g., marital conflict, unemployment and lack of social supports, can diminish parental psychological well-being which in turn negatively affects their parenting. This results in poor parent/child interaction and therefore negatively impacts on child development. Therefore, interventions targeting children and adolescent behavioral problems should aim to address risk and protective family factors systemically. Several studies have linked parental mental health to effectiveness of parenting interventions for children with severe conduct disorder. They suggested that although the positive change in parenting skills predicts an improvement in children outcomes, these treatment outcomes might be varied depending on parent’s ability to actively implement these acquired skills [27,28]. Parents who experience mental health issues, e.g., depression or anxiety might find it difficult to effectively implement these skills when distressed. Parental psychological well-being could be improved by adequate family, school and community supports which as a result are likely to promote parental competence. This is even more significant in families with disadvantaged backgrounds and/or minority groups [29,30,31,32]. More recent research has examined the mechanisms that contribute to the effectiveness of interventions for children and adolescent with conduct problems [13,14,28,33,34]. It suggested many parenting interventions that reported poor outcomes in disadvantaged families, often failed to encourage on-going engagement with families and service flexibility to overcome access barriers. The result from the meta-analysis examining the effectiveness of preventative interventions and treatments for youth antisocial behaviour, suggested that treatment approaches that actively engaged parents in the interventions such as a parent support group, a child-centered learning approaches and behavioural parenting training were associated with larger effects, and therefore were recommended when selecting effective interventions for youth with conduct disorders [35].

Multisystemic Therapy (MST) is an intensive family and community-based treatment targeting antisocial behaviours in adolescents (aged 11–16 years). The intervention is developed from the theory of social ecology introduced by Bronfenbrenner in 1979 [7] focusing on understanding multi-determined human behaviours by learning a complex interaction between individuals and various contextual influences within their life [7,36]. MST emphasizes the need to identify possible contributors to child’s behavioural problems both within systems and between systems in which the child is embedded. The probability of reducing antisocial behaviours will be increased by addressing these identified risk factors. MST intervention utilises a variety of evidence-based therapeutic treatments including behavioural parenting training (BPT), cognitive behaviour therapy, and structural family therapy, whilst employing family systems theory [37] and social ecological theories of behaviour [7]. MST is an intensive intervention, with a therapist having an average of 3 sessions per week in the family home for the duration of 4–5 months. A therapist has concurrent caseload of only 4 to 6 families; however, is available 24/7 to support parents in times of family disruption and distress during the intervention. The goals of the therapy are discussed and established at the early stage of the intervention by a therapist and the family members. Common family goals are reducing aggression, violence, and non-compliance in the home and community; improving school attendance and behaviour; and ceasing substance use and anti-social peer involvement. During MST treatment, parents work with a therapist to improve family functioning and their parenting skills such as monitoring skills, communication skills, problems solving skills, and emotional regulation. Therapists also liaise with schools and other services in the community to provide an on-going support for families if needed. Many research studies (including randomised controlled trials, case–control studies, cohort studies and benchmarking studies) have been conducted internationally by both MST model developers and independent researchers [38], and demonstrated that when the treatment model was implemented correctly (i.e., with high levels of prescribed treatment fidelity), the effectiveness of the intervention was high [39,40]. The findings from MST research indicate the parenting discipline approaches were found to be a key mediator of change for MST with adolescent conduct disorders [33,41]. An improvement in parental sense of competence during MST intervention contributes to positive changes in parental discipline, which result in improved parent–child relationship and decreased youth antisocial behaviour [42]. A multilevel meta-analysis that examined the impact of MST on youth with delinquency found significant treatment effects on delinquency, substance use, recidivism, family functioning and psychopathological symptoms. Although the finding noted that MST was most effective with delinquent youths under the age of 15, it suggested that an improvement of treatment for older youths may be achieved by focusing more on protective and risks factors in the peer group and school environment [43].

The majority of families referred to the Western Australian CAMHS MST service are socio-economically disadvantaged and experience a wide range of complex and challenging issues. These families have often experienced failed therapeutic interventions or had minimal positive contact with mental health and other social support services in the past. These families include many Australian Aboriginal families, and ethnoculturally and linguistically diverse (ELD) families. Over the past ten years, the WA CAMHS MST program has successfully engaged these at-risk populations and assisted them by re-engaging young people in educational/vocational settings, reducing youth homelessness, reducing or ceasing drug and alcohol use, and also preventing further involvement with the Police and Justice Departments. A critical initial goal of helping these populations is developing a strong working alliance with parents and other caregivers [44]. The program excels in achieving this critical initial stage of the intervention by working with families in their homes and communities. The local study conducted within the Westertn Australian Child and Adolescent Mental Health Service (CAMHS) [45], found favourable and enduring outcomes for most families completing the MST intervention.

Previous studies indicated that interventions which address risk and protective factors using systemic approaches were more likely to prove effective for children and adolescents with disruptive behaviours. Nevertheless, the understanding of parental factors as moderators of the success of interventions still requires further exploration. Therefore, the aims of this retrospective study were to examine changes in adolescent behaviours, and parental factors such as parental mental health, discipline approaches and monitoring skill following the MST intervention. Secondly, we aimed to determine the mediating role that parental discipline approaches and monitoring skill have in the relationship between parental mental health and adolescent behavioural problems. The mediating effects were tested at post-treatment because we sought to examine how parent’s depression, anxiety and stress level could affect parent’s ability to implement acquired parenting skills immediately after the intervention. We hypothesised that (1) adolescent behavioural problems, parental depression, anxiety and stress, parental discipline approaches and monitoring skill would improve after the treatment and be sustained at the 12-month follow-up. (2) at post-treatment caregivers who presented with lower level of anxiety, stress and depression would be more likely to report higher level of monitoring skills and lower level of authoritarianism (described as hostile and low warmth), or permissiveness in their parenting approach. Consequently, these improved parenting skills would contribute to a decrease in adolescent behavioural problems. We hope information gained from this study is used to inform programs and practitioners working with disadvantage families, in order to understand the mechanisms that may impact the effectiveness of interventions.

## 2. Method

### 2.1. Participants and Procedure

This retrospective study extracted the data collected from 193 families engaged with the MST research program during 2014–2019. Families were assured their decision to participate in the research was voluntary, and that they could withdraw at any time. Once families agreed to participate in the research, written informed consent was obtained from caregivers. Then, caregivers were contacted by research staff to schedule a face-to-face interview. Instruments used for the data collection contained face-to-face interviews and questionnaires which were collected at baseline, post-treatment, and 6 & 12 month follow up. The data collection was approved by the Department of Health, Human Research Ethics Committee (DoH, HREC), Western Australia.

### 2.2. Measures

#### 2.2.1. Child Behaviour Checklist (CBCL)

CBCL assesses child behaviours and competencies in the context of psychopathology, and in this study the parent-reported version was administered to monitor changes in children’s behaviours over time. Caregivers rated childhood internalising behaviours (e.g., anxious/depressed, withdrawn, somatic complaint), externalising behaviours (e.g., rule-breaking behaviour and aggressive behaviour), social problems, thought problems, attention problems and other behavioural problems. It consists of 113 items scored on a 3-point Likert scale: not at all (0), somewhat true (1) and very true (2). The scale has high psychometric properties with internal reliability (Chronbach’s α) of 0.97 for total empirically based problem scales and the alphas of each subscales ranging from 0.79 to 0.97 [46].

#### 2.2.2. Depression, Anxiety and Stress Scale-21 (DASS-21)

DASS-21 was a self-report scale completed by caregivers to measures their negative emotional states of depression, anxiety and stress. DASS-21 used for the purpose of this research is an abbreviated version with three subscales (depression, anxiety and stress) of 7 items each. The internal reliability for the standardised 7-item scales is 0.81 for depression, 0.73 for anxiety and 0.81 for stress [47]. The total score for each subscale is determined by combining the scores of the 7 corresponding items and multiplying it by 2. An increase in subscale score(s) over a period of time indicates deterioration in the caregiver’s mental health. These scores were also used to determine the level of severity of the caregiver’s depression, anxiety and stress. Originally, the level of severity is categorised into normal, mild, moderate, severe or extremely severe. However, for the purpose of this study researcher re-categorised them into two subgroups: non-clinical range (i.e., normal and mild) and clinical range (i.e., moderate, severe and extremely severe).

#### 2.2.3. Parenting Styles and Dimensions Questionnaire (PSDQ)

PSDQ was reported by caregivers [48] and used to measure parenting discipline approaches along a continuum of Baumrind’s Typology of authoritative, authoritarian, and permissive parenting styles [49]. The PSDQ contains 32 statements describing different caregiver’s responses to a child’s behaviour. It has a 5-point scale ranging from ‘never’ to ‘always’ to rate the frequency of certain discipline approaches and responses used by the caregiver. The statements cover three dimensions of authoritative approach (connection, regulation and autonomy) with internal reliability of 0.86, three dimensions of authoritarian approach (physical coercion, verbal hostility and non-reasoning/punitive) with internal reliability of 0.82 and one dimension of permissiveness (indulgence) with internal reliability of 0.64. For the purpose of this study, only the scores of authoritarian and permissive discipline approaches were observed. The decreased scores of authoritarian and permissive parenting approaches over a period of time indicate a reduction in a caregiver’s negative discipline approach.

#### 2.2.4. Parental Monitoring Scale

Parental Monitoring scale was adapted from an existing scale developed by Stattin and Kerr [50], which includes 8 questions using 5-point Likert scales ranging from never to always. This self-report scale asks caregiver about knowledge of their child’s whereabouts, activities, and associations (e.g., “How often do you know: what your child is doing during their free time? with whom your child is spending their free time? what your child spends their money on?”). The internal reliability for this adapted 8-item parental monitoring scale was 0.88. The increased score of parental monitoring over period of time indicate improvement in caregiver’s monitoring skill.

### 2.3. Data Analytic Strategy

Extracted data were analysed using statistic software SPSS for Window version 24 (IBM Corp., Armonk, NY, USA). Socio-demographic data were analysed using descriptive statistics for continuous numerical variables and absolute and relative frequencies for nominal qualitative variables. Due to some missing scores in follow-up data, multiple imputation was performed as a method for handling missing data as recommended by Van Ginkel et al. [51]. They suggested that multiple imputation was an optimal method providing a solution for problems that commonly found in those traditional methods of handling missing data (i.e., listwise deletion, pairwise deletion, and (single) imputation). The problems such as wastefulness, computational problems, biased (co)variances, and biased p values and confidence intervals could be addressed using the statistical model that accurately describes the data and its random error component in order to create several plausible complete versions of the incomplete data sets. Multiple different outcomes are produced as a result of using different version of complete data sets in statistical analyses and these outcomes are combined into an overall statistical analysis in which the standard errors and significance tests were employed.

To test the hypotheses, we have broken down the analyses into three stages. For the first hypothesis, the preliminary analyses were performed as a stage one to investigate the number of adolescents with improved behaviours at post-treatment and follow-ups and to examine the number of parents reporting the clinical range in depression, anxiety and stress at different time points. To determine range of change in adolescent behaviours (CBCL) from baseline, the value of ±0.5 of one standard deviation was used as an index of significant change as recommended by Key Performance Indicators for Australian Public Mental Health Services [52]. Adolescents with follow-up scores increased from baseline more than 0.5 SD was considered as ‘deteriorate’, maintained within ±0.5 SD as ‘no change’, and scores that decreased more than −0.5 SD as ‘improvement’.

For stage two, the long-term outcomes were investigated by using one-way repeated measures ANOVA. These analyses were applied to investigate the change of adolescent and parental outcomes at different time points. At the beginning of analysis, an assumption testing for normality, homogeneity of variance and sphericity was conducted. The severity of departures from sphericity in one-way repeated-measures ANOVA was assessed by using Mauchly’s test. If a statistical significance in Mauchly’s test was detected, it indicated that there was a significant difference between the variances and the assumption of sphericity was violated for the main effects [53]. As a result, the obtained F-ratio was evaluated using new degree of freedom, which are calculated using the less conservative correction called Huynh-Feldt Epsilon [54,55].

The scores at baseline were compared with the scores at post-treatment/follow-ups. A significant difference existing between baseline and post-treatment/follow-ups demonstrated changes in adolescent behavioral problems, caregiver’s mental health, parental discipline approaches and monitoring skill. A Partial Eta Squared (*η*^2^*_p_*) is a measure of effect size from the main ANOVA which could be obtained from SPSS output (as reported in Tests of within-Subjects Effects table). However, an effect size (*r*) for a pair comparison should also be reported in addition to the main ANOVA as recommended by Field [54]. Therefore, we also calculated an effect size for its’ contrasts by which *F*-values were converted to *r*. An equation used for calculating is as follows:(1)$$r=\sqrt{F\left(1,df_{R}\right)/\left(F\left(1,df_{R}\right)+df_{R}\right)}$$

Cohen [56] reported the following intervals for *r*: 0.1 to 0.3 as small effect; 0.3 to 0.5 as intermediate effect; 0.5 and higher as strong effect.

For the second hypothesis, the mediating effects of parenting discipline approaches and monitoring skill on the relationship between parental mental health and adolescent behavioural problems were examined as the third stage of analyses. The parallel multiple mediator model was performed using PROCESS V3.4 Macro for SPSS developed by Andrew F. Hayes [57]. Firstly, the correlation analysis was performed to examine the inter-correlation between all variables. Then, we tested the hypothesis that at the post-treatment the relationship between parental mental health (i.e., depression (*X*_1_), anxiety (*X*_2_) and stress (*X*_3_)) and adolescent behavioural problems (*Y*) would be mediated by authoritarian approach (*M*_1_), permissiveness (*M*_2_) and monitoring skill (*M*_3_). The scores from post-treatment were used in these analyses because researchers aimed to examine the mediating effects after the families had received interventions from MST. The aim was to determine how parent’s depression, anxiety and stress level could affect parent’s ability to implement acquired parenting skills immediately after the intervention, which in turn possibly resulted in varied adolescent behavioural outcomes.

Figure 1 depicts a process in which the independent variables led to the mediators and the mediators then led to the dependent variable. With *k* = 3 mediators, four equations are needed:*M*_1_ = *i_M_*_1_ + *a*_1_*X* + *e_M_*_1_
*M*_2_ = *i_M_*_2_ + *a*_2_*X* + *e_M_*_2_
*M*_3_ = *i_M_*_3_ + *a*_3_*X* + *e_M_*_3_
*Y* = *i_Y_* + *c’X* + *b*_1_*M*_1_ + *b*_2_*M*_2 +_ *b*_3_*M*_3_ + *e_Y_*(2)

## 3. Results

### 3.1. Descriptive Statistic

A total of *n* = 193 families were included in the analysis, and 73% (*n* = 141) of adolescents were male. The mean age of adolescents was 13.7 years (SD = 1.40, range 11–16 years). Majority of adolescents were identified as Caucasian (85%), 8% as ethnoculturally and linguistically diverse (ELD) and 7% as Australian Aboriginal. Around half of these adolescents (51%) lived with a single caregiver, 23% with an intact family, 20% with a blended family, and 6% lived with caregivers who were not biological parents (e.g., foster parents, grandparents or relatives). Around half of caregivers had a high school education or lower (53%), and 53% of families had an annual income not included welfare benefits < A$50,000 per annum. Around half of these adolescents had used illicit drugs or alcohol at least once in the previous 6 months. 90% (*n* = 174) of parents who participated in the research were female which included biological mothers, stepmothers, foster mothers and grandmothers.

### 3.2. Changes in Adolescent Behaviours and Parental Factors at Different Timepoints

#### 3.2.1. Preliminary Finding

The result from preliminary analyses using an index of significant change demonstrates that 80% of adolescents exhibit an improvement in total behaviours at the post-treatment and at following 6 and 12 months after the MST intervention (Table 1). The result from parental DASS (Figure 2) indicates that at baseline around half of caregivers reported their stress, anxiety and depression in the clinical range. However, these numbers decrease after the MST intervention, and continue to decrease at 6- and 12-month follow-up.

#### 3.2.2. Long-Term Outcomes Finding

The long-term outcomes in adolescents and caregivers were examined using the repeated measure ANOVA. The results from the repeated measure ANOVA (Table 2) indicated that CBCL internalising problems, externalising problems and total problems scores decreased significantly over time as follows: internalising problems, *F*(2.60, 499.13) = 90.47, *p* < 0.001, *η*^2^*_p_* = 0.32; externalising problems, *F*(2.86, 549.37) = 203.87, *p* < 0.001, *η^2^_p_* = 0.52 and total problems, *F*(2.76, 529.54) = 206.98, *p* < 0.001, *η^2^_p_* = 0.52. Secondly, the parental mental health also had improved over time as follows: depression, *F*(2.73, 524.73) = 30.82, *p* < 0.001, *η^2^_p_* = 0.138; anxiety, *F*(2.63, 503.97) = 17.26, *p* < 0.001, *η^2^_p_* = 0.08; and stress, *F*(2.79, 534.91) = 63.27, *p* < 0.001, *η*^2^*_p_* = 0.25. Thirdly, the authoritarian and permissive parenting style scores had decreased over time as follows: authoritarian parenting style, *F*(2.50, 479.41) = 165.69, *p* < 0.001, *η*^2^*_p_* = 0.463; and permissive parenting style, *F*(2.61, 500.08) = 161.58, *p* < 0.001, *η^2^_p_* = 0.457. Lastly, the parental motoring skill had increased over time *F*(2.46, 470.40) = 16.29, *p* < 0.001, *η*^2^*_p_* = 0.079.

A series of pair-wise comparisons demonstrated that there were statistically significant differences between baseline scores and follow-up scores with medium to large effect sizes found between baselines vs. post-treatment/follow-ups in most measures. The results confirm enduring positive adolescent and parental outcome scores at post-treatment and at follow-up times.

### 3.3. Mediating Effect Findings

The inter-correlation between each of the parental factors (i.e., parental depression, anxiety, stress, authoritarian, permissiveness and monitoring) and adolescent behavioural problems at post-treatment were detected and found to be all statistically significant. The parallel multiple mediator models (Table 3 and Figure 3) illustrated the total (*c*), direct (*c’*) and indirect (*a_i_b_i_*) effects of parental mental health on adolescent behavioural problems with parental approaches and monitoring skill as mediating variables. The indirect effects of parental anxiety on adolescent behavioural problems through mediating variables were estimated (*a_i_b_i_*) as follows: authoritarian = 0.039, permissiveness = 0.374 and monitoring skill = 0.155. The indirect effects of stress on adolescent behavioural problems through mediating variables were estimated as follows: authoritarian = 0.005, permissiveness = 0.259 and monitoring skill = 0.157. The indirect effects of depression on adolescent behavioural problems through mediating variables were estimated as follows: authoritarian = 0.059, permissiveness = 0.240 and monitoring skill = 0.186.

Around a third of the variance in adolescent behavioural problems was accounted for by proposed mediators (i.e., discipline approaches and monitoring skill) and parental stress, anxiety and depression. The indirect effect pathways indicated there were significant associations found between parental mental health (i.e., anxiety, stress and depression) and parental authoritarian, permissiveness and monitoring skill (path *a*_1,_ *a*_2_, *a*_3_). Parental authoritarian approach was predicted by anxiety (*R*^2^ = 0.21), stress (*R*^2^ = 0.16) and depression (*R*^2^ = 0.12). Parental permissiveness was predicted by anxiety (*R*^2^ = 0.15), stress (*R*^2^ = 0.12) and depression (*R*^2^ = 0.15). Parental monitoring skill was slightly and negatively predicted by anxiety (*R*^2^ = 0.02), stress (*R*^2^ = 0.03) and Depression (*R*^2^ = 0.06). Adolescent behavioural problems were found to be positively predicted by parental permissiveness whereas negatively predicted by monitoring skill. Parental authoritarian was slightly associated with adolescent behavioral problems; however, it was not statistically significant.

## 4. Discussion

The first aim of this study was to observe the changes in adolescent and parental outcomes after the MST intervention. The results of the preliminary and longitudinal data analysis supported the first hypothesis which indicated that the majority of adolescents referred to MST exhibited positive changes in their emotional and behavioural problems post-treatment, and these changes were sustained over the following 12 months period. The results also indicated the majority of caregivers reported significant and enduring improvement in their mental health, parenting and monitoring skills after their involvement with the MST intervention. This retrospective study indicated that Multisystemic Therapy, had an enduring positive impact on adolescents and their families. The desired outcomes of the treatment were achieved by increasing caregiver capacity to implement effective parenting skills with the aim of successfully eliciting positive behaviours in their child.

The second aim of this study was to determine the mediating role that parental discipline approaches and monitoring skill have in the association between parental mental health and adolescent behavioural problems. The results from the mediation analysis confirmed both direct and indirect effects between parental mental health and adolescent behavioural problems as indicated in the second hypothesis. The indirect associations demonstrated that parental anxiety, stress and depression predicted authoritarian approach, permissiveness and adversely predicted parental monitoring skill. Subsequently, parental authoritarian approach, permissiveness and monitoring skill were found to associate with adolescent behavioural problems. The results suggested that caregivers who reported high level of depression, anxiety or stress were more likely to report high level of authoritarian or permissive approach, and poor parental monitoring skill, which as a result contributed to more problem behaviours in adolescents. It is also worth noting that when comparing all three mediating variables, parental permissiveness was deemed to be the strongest predictor of adolescent behavioural problems.

These findings are consistent with other studies [11,21], suggesting that caregivers with poor mental health were more likely to use negative parenting styles (e.g., physical punishment, verbal hostility and/or avoidance), compared to caregivers with better mental health. Therefore, the intervention that improves caregiver’s mental well-being would likely enhance their parenting skills, which in turn would promote adolescent positive outcomes. As recommended by previous research, obtaining positive changes in caregiver’s parenting skills and mental health are strong indicators for positive treatment outcomes [58,59,60]. This supports the notion that for practitioners to provide an effective intervention for parents supporting their adolescents with emotional and behavioural difficulties, parental mental health issues should also be addressed. Our findings from the mediation analyses suggest that parents who reported low level of depression, anxiety and stress at post-treatment are more likely to effectively implement positive parenting skills acquired from the treatment which improve their child’s behaviours and general functioning.

The MST intervention places a focus on teaching caregivers improved communication skills, and effective techniques to manage anti-social behaviours and elicit pro-social behaviours in their children. Therefore, the caregiver’s ability to implement these skills needs to be evaluated and discussed throughout the intervention. When mental health is found to be a barrier for caregivers to be consistent with their parenting; it is important that the clinician address this issue. The therapeutic relationship between the clinician and caregivers increases positive engagement with the program and encourages caregivers to seek on-going support to improve their own mental well-being. The outcome from this study confirms previous studies findings that with the right combination of family and social support, caregivers with mental health issues can improve their own mental well-being, learn to parent well and enrich relationships with their children [1,20,21].

There are some methodological limitations that must be taken into consideration when interpreting the results of this study. Without a control or comparison group, it is difficult to exclude the possible confounding impact of natural variations over time. The results also indicated that only around a third of the variance was accounted for by proposed mediators and parental metal health factors. Given that this is a retrospective study, researchers had limited information on other risk factors such as historical family trauma, domestic violence, individual learning disability and/or cognitive impairment, etc. Therefore, this study did not have an opportunity to explore these factors and how they are correlated. This suggested that other confounding factors contributing to adolescent behavioural problems should be further investigated. Despite this limitation, this study provides substantial evidence indicating caregiver mental well-being and positive parenting discipline approaches influence positive outcomes with adolescents. Previous researchers who have examined the evidence-base for MST [61,62] noted that with many existing randomised controlled trials demonstrating the effectiveness of the MST model (Multisystemic Therapy: Research at a glance, 2022) [38] further research should focus on elucidating the underlying mechanisms of MST effectiveness. Understanding the enhancing factors of intervention effectiveness, is important for planning policy and clinical guidelines [1].

Another limitation of this study is all the instruments used for this study were based on parental ratings, either for the adolescent (CBCL) or parent (DASS21, PSDQ and parental monitoring scale). Inclusion of multi-informant measures, e.g., child self-reported and teacher-reported would provide more perspectives for a comprehensive examination. A longer follow-up period is also recommended to confirm current results. Further evaluation may include a descriptive analysis of family historical and environmental factors, analysis of comparison groups including cost–benefit analyses, and examination of other confounding factors that potentially contribute to the successful implementation of the MST intervention.

## 5. Conclusions

Effective intervention with high-risk youth having major behavioural issues has the potential to positively alter the life-trajectory of these young individuals, and avoid predictable negative outcomes including chronic adult unemployment, patterns of inter-personal aggression, family and domestic violence, various mental illness, substance abuse, anti-social and criminal behaviour, probable periods of youth and adult incarceration, and premature death. Effective parent interventions within these families typically involves teaching parents and caregivers improved communication and problem-solving skills designed for generalisation and possible use with any other children having chronic behavioural difficulties. The MST intervention therefore has potential for powerful and enduring positive social influence, resulting in significant cost-saving potential for the wider community across numerous domains of influence mentioned.

## Figures and Tables

**Figure 1 ijerph-19-13418-f001:**
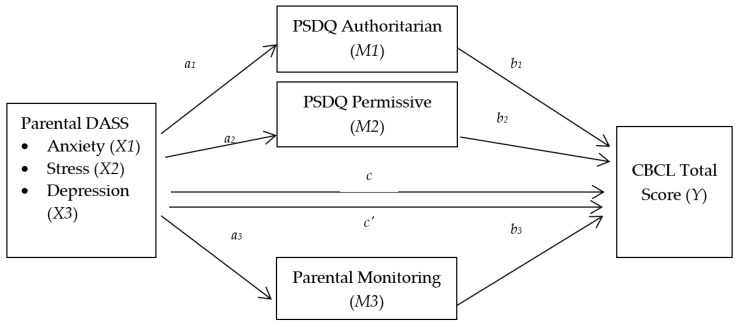
A conceptual diagram of the parallel multiple mediator model.

**Figure 2 ijerph-19-13418-f002:**
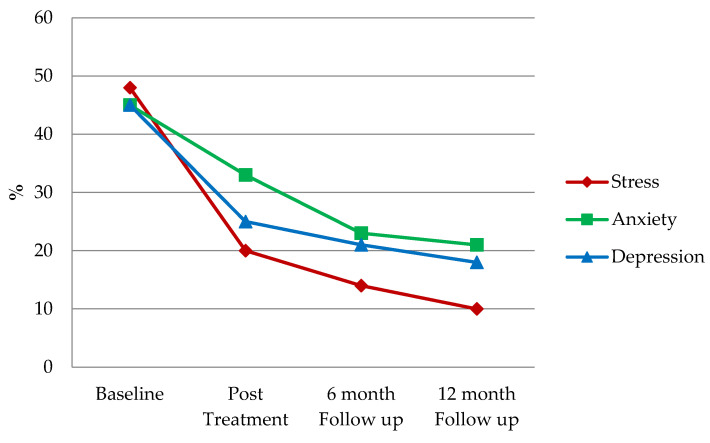
% of caregivers who reported DASS in clinical range at baseline, post-treatment, 6-month and 12 month follow-up (*n* = 193).

**Figure 3 ijerph-19-13418-f003:**
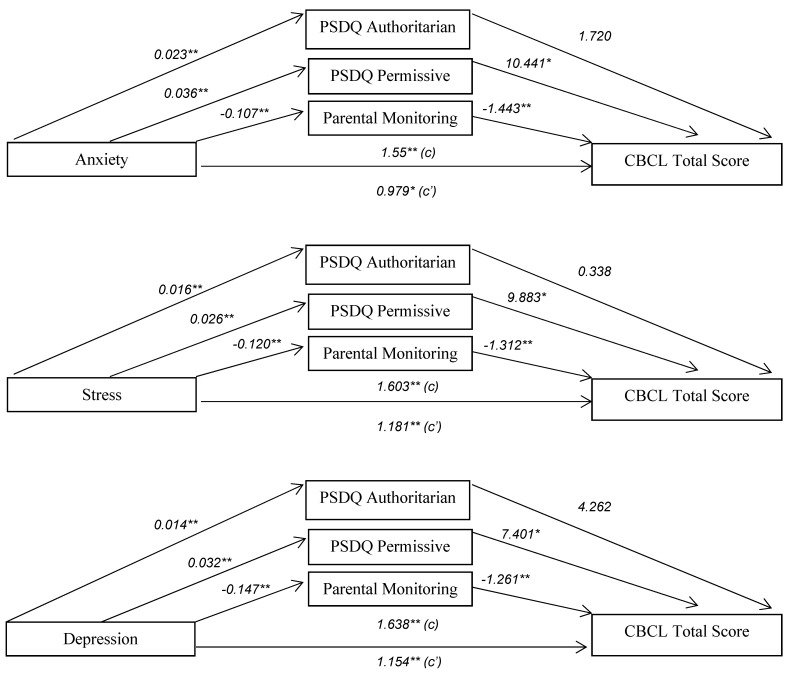
Parental discipline approaches and monitoring skill as mediators of the relationship between parental mental health and adolescent behavioural problems. Note: *c* = total effect, *c’* = direct effect, * *p <* 0.05, ** *p* < 0.01.

**Table 1 ijerph-19-13418-t001:** % of adolescents in different categories of change from baseline in CBCL at post-treatment, 6- and 12- month follow-up (*n* = 193).

CBCL		Categories of Change					
		Improvement		No Change		Deteriorate	
		*n*	%	*n*	%	*n*	%
	Post	117	60.6	69	35.8	7	3.6
Internalising	6-month	116	60.1	63	32.6	14	7.3
	12-month	126	65.3	52	26.9	15	7.8
	Post	152	78.8	36	18.6	5	2.6
Externalising	6-month	160	82.9	25	13.0	8	4.1
	12-month	168	87.1	18	9.3	7	3.6
	Post	151	78.2	37	19.2	5	2.6
Total	6-month	155	80.3	32	16.6	6	3.1
	12-month	165	85.5	23	11.9	5	2.6

**Table 2 ijerph-19-13418-t002:** CBCL, Parental mental health, Parental monitoring & Parenting Styles outcome measure: Summary of long-term results for MST research participants.

x	Time (*n* = 193), Mean (SD)				Estimated Mean Differences					
	Baseline	Post-Treatment	6-Month Follow-Up	12-Month Follow-Up	Baseline to Post-Treatment		Baseline to 6-Month Follow-Up		Baseline to 12-Month Follow-Up	
					Mean Difference (95% CI) *p*	Effect Size *r* (90% CI)	Mean Difference (95% CI) *p*	Effect Size *r* (90% IC)	Mean Difference (95% CI) *p*	Effect Size *r* (90% CI)
CBCL										
Internalising	21.69 (11.58)	13.18 (9.30)	12.20 (9.19)	9.80 (8.45)	8.50 (6.60 to10.40) <0.001	0.65 (0.58 to 0.70)	9.48 (7.30 to 11.67) <0.001	0.64 (0.57 to 0.69)	11.88 (9.35 to 14.41) <0.001	0.67 (0.60 to 0.71)
Externalising	18.49 (10.33)	11.98 (8.97)	9.75 (8.18)	9.85 (7.51)	6.51 (4.62 to 8.40) <0.001	0.55 (0.47 to 0.62)	8.75 (6.73 to 10.76) <0.001	0.64 (0.57 to 0.69)	8.64 (6.47 to10.81) <0.001	0.61 (0.53 to 0.67)
Total	93.04 (29.18)	52.25 (33.39)	42.71 (31.98)	35.52 (27.88)	40.79 (34.47 to 47.12) <0.001	0.78 (0.73 to 0.81)	50.33 (43.00 to 57.66) <0.001	0.80 (0.75 to 0.83)	57.52 (50.53 to 64.50) <0.001	0.85 (0.81 to 0.87)
Parental MH										
Depression	13.68 (11.37)	8.70 (9.15)	8.74 (8.00)	6.93 (6.56)	4.98 (3.02 to 6.94) <0.001	0.44 (0.35 to 0.52)	4.93 (2.78 to7.09) <0.001	0.40 (0.30 to 0.49)	6.74 (4.56 to 8.93) <0.001	0.51 (0.42 to 0.58)
Anxiety	10.01 (9.65)	6.79 (7.31)	6.25 (6.53)	5.95 (5.84)	3.22 (1.59 to 4.85) <0.001	0.35 (0.24 to 0.45)	3.77 (1.90 to 5.63) <0.001	0.36 (0.26 to 0.46)	4.06 (2.06 to 6.07) <0.001	0.36 (0.26 to 0.46)
Stress	18.49 (10.33)	11.98 (8.97)	9.75 (8.18)	9.85 (7.51)	6.51 (4.62 to 8.40) <0.001	0.55 (0.47 to 0.62)	8.75 (6.73 to 10.76) <0.001	0.64 (0.57 to 0.69)	8.64 (6.47 to 10.81) <0.001	0.61 (0.53 to 0.67)
Parental Monitoring	28.00 (6.85)	31.00 (5.84)	29.96 (6.44)	29.85 (6.10)	−3.00 (−4.19 to −1.82) <0.001	0.44 (0.33 to 0.52)	−1.96 (−3.38 to −0.53) <0.05	0.26 (0.14 to 0.36)	−1.85 (−3.38 to −0.51) <0.05	0.26 (0.14 to 0.36)
Authoritarian										
Total	1.95 (0.55)	1.33 (0.67)	1.04 (0.71)	0.821 (0.69)	0.63 (0.49 to 0.76) <0.001	0.68 (0.61 to.72)	0.91 (0.75 to 1.07) <0.001	0.74 (0.69 to 0.77)	1.13 (0.96 to 1.30) <0.001	0.79 (0.75 to 0.82)
Permissive										
Indulgent	2.84 (0.84)	1.86 (0.95)	1.53 (1.05)	1.20 (0.97)	0.99 (0.85 to 1.12) <0.001	0.72 (0.66 to 0.76)	1.31 (1.14 to 1.48) <0.001	0.74 (0.69 to 0.78)	1.64 (1.46 to 1.82) <0.001	0.80 (0.75 to 0.83)

**Table 3 ijerph-19-13418-t003:** Regression coefficients, standard errors, and model summary information for the parallel multiple mediator model (*n* = 180).

Antecedent	Consequent															
		*M*_1_ (Authoritarian)				*M*_2_ (Permissive)				*M*_3_ (Monitoring)				*Y* (CBCL Total)		
		Coeff.	SE	*p*		Coeff.	SE	*p*		Coeff.	SE	*p*		Coeff.	SE	*p*
*X*_1_ (Anxiety)	*a* _1_	0.023	0.003	<0.001	*a* _2_	0.036	0.006	<0.001	*a* _3_	−0.107	0.059	0.071	*c’*	0.979	0.300	0.001
*M*_1_ (Auth)		-	-	-		-	-	-		-	-	-	*b* _1_	1.720	6.559	0.794
*M*_1_ (Perm)		-	-	-		-	-	-		-	-	-	*b* _2_	10.441	3.493	0.003
*M*_1_ (Moni)		-	-	-		-	-	-		-	-	-	*b* _3_	−1.443	0.348	<0.001
Constant	*i_M_* _1_	1.469	0.003	<0.001	*i_M_* _2_	2.01	0.063	<0.001	*i_M_* _3_	32.226	0.579	<0.001	*i_Y_*	75.26	16.416	<0.001
		*R*^2^ = 0.212 *F*(1, 178) = 47.780 *p* < 0.001				*R*^2^ = 0.149 *F*(1, 178) = 31.172 *p* < 0.001				*R*^2^ = 0.018 *F*(1, 178) = 3.312 *p* = 0.071				*R*^2^ = 0.301 *F*(4, 175) = 18.854 *p* < 0.001		
*X*_2_ (Stress)	*a* _1_	0.016	0.003	<0.001	*a* _2_	0.026	0.005	<.001	*a* _3_	−0.120	0.048	0.013	*c’*	1.181	0.227	0.001
*M*_1_ (Auth)		-	-	-		-	-	-		-	-	-	*b* _1_	0.338	6.157	0.956
*M*_1_ (Perm)		-	-	-		-	-	-		-	-	-	*b* _2_	9.883	3.331	0.003
*M*_1_ (Moni)		-	-	-		-	-	-		-	-	-	*b* _3_	−1.312	0.335	<0.001
Constant	*i_M_* _1_	1.469	0.003	<0.001	*i_M_* _2_	2.01	0.063	<0.001	*i_M_* _3_	32.226	0.579	<0.001	*i_Y_*	65.45	15.466	<0.001
		*R*^2^ = 0.155 *F*(1, 178) = 32.620 *p* < 0.001				*R*^2^ = 0.119 *F*(1, 178) = 24.144 *p* < 0.001				*R*^2^ = 0.034 *F*(1, 178) = 6.314 *p* <0.05				*R*^2^ = 0.358 *F*(4, 175) = 24.386 *p* < 0.001		
*X*_3_ (Depression)	*a* _1_	0.014	0.003	<0.001	*a* _2_	0.032	0.005	<0.001	*a* _3_	−0.147	0.046	0.002	*c’*	1.153	0.226	<0.001
*M*_1_ (Auth)		-	-	-		-	-	-		-	-	-	*b* _1_	4.262	6.022	0.480
*M*_1_ (Perm)		-	-	-		-	-	-		-	-	-	*b* _2_	7.401	3.450	0.033
*M*_1_ (Moni)		-	-	-		-	-	-		-	-	-	*b* _3_	−1.261	0.337	<0.001
Constant	*i_M_* _1_	1.489	0.037	<0.001	*i_M_* _2_	1.942	0.065	<0.001	*i_M_* _3_	32.885	0.602	<0.001	*i_Y_*	67.95	15.515	<0.001
		*R*^2^ = 0.120 *F*(1, 178) = 24.359 *p* < 0.001				*R*^2^ = 0.149 *F*(1, 178) = 42.718 *p* < 0.001				*R*^2^ = 0.055 *F*(1, 178) = 10.300 *p* <.001				*R*^2^ = 0.355 *F*(4, 175) = 24.033 *p* < 0.001		

## Data Availability

The data that support the findings of this study are the property of Department of Health, Western Australia therefore are not publicly available due to their containing information that could compromise the privacy of research participants. As a result, non-identifiable data are only available from the corresponding author upon reasonable request.

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
