# Peer review of "Mediating Effects of Discipline Approaches on the Relationship between Parental Mental Health and Adolescent Antisocial Behaviours: Retrospective Study of a Multisystemic Therapy Intervention"

_ijerph, 2022, doi:10.3390/ijerph192013418_

Round 1

Reviewer 1 Report

This is paper focusses on examining changes in parenting and adolescent behaviour following MST in single group repeated measures study design. It also examines the mediating role of parental discipline in the relationship between parent mental health and adolescent behaviour. The paper has a number of strengths, including the focus on behaviour problems in adolescents, which is frequently neglected group for intervention research. It also includes a good sample size, appropriate and validated measures, and long-term follow-up of the sample up to 12 months post intervention. Unfortunately, however, there are a number of weaknesses with the study and the paper. The key weaknesses include: (1) the need for more literature on previous MST studies and mediating role of discipline in adolescent samples, (2) clearer aims and hypotheses and a statement regarding the novelty of the research (how it addresses gaps in previous research); and (3) a clearer description of the analysis conducted, as presently the analyses and results are somewhat difficult to follow and confusing for the reader. Key points are addressed in turn below.

Title

1. The title of the paper is quite long; consider shortening

Abstract

 2. The finding of the analyses should be expressed in terms of statistically significant improvements rather than broad statements like “the majority of adolescents and parents exhibited positive changes.”

Introduction

3.  The introduction requires further details of the extant research on MST with adolescents and also research on mediating role of discipline. It must be clear to the reader what are the gaps in research on these topics to date, and how the current research study addresses these gaps.

4.  The first sentence of the introduction states that ODD/CD are the most common disorders in childhood, but more recent data suggests that they are second to anxiety. (see worldwide prevalence data in Polanczyk et al., 2015)

5. Consider moving the paragraph on WA CAMHS to the end of the introduction to improve flow.

6. The sentence on “We hope information gained from this study…” should be moved to after the stated aims for the study.

7. The aims for the study need to be reworded (e.g., “determine enduring positive changes in adolescent’s behaviour…” change to something like: examine changes in..). It is important that clearly worded and numbered hypotheses are included, and there is sufficient literature cited in the introduction to support these hypotheses. The hypotheses should also refer to the timepoints examined, including post, and follow-up timepoints. It is currently unclear how the mediation effect is being explored within the design of the longitudinal study, so further information is needed on this point (more on this below).

Method

8. Report Cronbach’s alpha with zero before decimal point. This is also referred to by different terms throughout such as “reliability”, “internal consistency” and “internal reliability” so keep terms consistent.

9. It is unclear why a 0.5 of one SD is used an index of significant change. This requires a reference but should go in the data analytic plan section, not the measures section.

10.  The data analytic plan should clearly state the analysis used to answer each hypothesis, to assist the reader to understand and follow. At present, it is not clear how the first hypothesis is being addressed. Is it through repeated measures ANOVA or through looking at categories of change (is this clinically significant change?) or both? Given the analysis of categories of change does not include statistical analyses of significance, it makes sense to present the ANOVA results first and foremost. It is also unclear how the second hypotheses in relation to the mediation analyses is being addressed. It is unclear which timepoints are being examined in the mediation model. If all variables are examined at post (I am unsure if this is the case), give a rationale for why this is being done.

Results

11. At present the results section is confusing for the reader as it is not clear how different analyses are being used to answer the hypotheses in the study. This section needs reworking to match the data analytic plan.

12. The sub-heading 3.1 labelled ‘preliminary results’ is incorrect as it describes the sample. Also the sub-heading 3.2 on long-term findings describes both short- (i.e., post-treatment) and long-term findings, so this needs rewording.

13. Report on the gender of the participating parent (mothers vs father)

Discussion 

14. Ensure that the discussion is structured according to the hypotheses presented in the Introduction and clearly state whether or not support was found for each of the hypotheses.

15. Given the confusion with the analyses, it is difficult to comment on the interpretation of findings, particularly in relation to the mediation analysis. For example, the statement that “ ..findings indicate that improved mental well-being had a positive impact on parenting skills which improves their child’s behaviour and general functioning” (line 46-48); however it is not clear how ‘improved mental well-being’ is assessed. Did you examine changes in parental mental well-being as a variable, or are you assuming changes have occurred because the post-intervention score on this measure is being used?

General

16. There are typos and errors throughout (particularly in punctuation, such as two full stops, and also in spacing, such as extra spaces). The manuscript requires a thorough proof read with corrections made. e.g., line 54, 55, 57, 150, 186 etc

Author Response

Thank you so much for your valuable feedbacks on our manuscript. Please find attached file which indicates our responses to your comments.

Reviewer 2 Report

I welcome this study because of the relevance that its conclusions may bring to future intervention programes.  I am going to write here some comments hopefully useful to you.

I suggest to improve the introduction. It would be important to better motivate the choice of the variables considered (anxiety, stress and depression; discipline approaches and monitoring skill), citing the bibliographic references too.

In addition to the dimensions of anxiety, stress and depression, it would be useful to consider other variables that the literature identifies as related to children antisocial behavior: family conflicts, violent behaviors by adults towards other family members, psychological or physical abuse.

It would be useful to verify the variation of these variables after the intervention.

In the introduction (lines 90-97), some of these variables are mentioned but not actually considered in the study. A motivation of the choice of the variables under study and the exclusion criteria, would be useful, even necessary, both supported by the scientific literature.

I propose also to briefly motivate the absence of risk factor referred to school, for example learning diseases and scholastic failure.

I think that you, rightly, took care of data analysis. Advanced statistical analysis, such as the one used in the article, is not a usual tool of my work, then I do not feel competent in evaluating the strategy widely described in paragraghs/chapters ranging from paragraph 2.3 to 3.3.

I appreciate the clarity in the presentation of the discussions (chapter 4) and agree with the limitations of the study as described by the authors. 

Author Response

Thank you so much for your valuable feedbacks on our manuscript. Please find attached document which indicates our responses to your comments.

Reviewer 3 Report

The author has done a fantastic study on the Relationship between parental mental health and adolescent antisocial behaviour. The approach is nice. The author should come out with an annexure with the scale item used in this study so that other scholars will be really helpful to them when they refer to it.  

Author Response

Thank you so much for your feedback on our manuscript. Please find attached document which indicates our response to your comment.
